# Unsupervised Learning of Compositional Energy Concepts

**Yilun Du**
MIT CSAIL
yilundu@mit.edu

**Shuang Li**
MIT CSAIL
lishuang@mit.edu

**Yash Sharma**
University of Tübingen
yash.sharma@uni-tuebingen.de

**Joshua B. Tenenbaum**
MIT CSAIL, BCS, CBMM
jbt@mit.edu

**Igor Mordatch**
Google Brain
imordatch@google.com

## Abstract

Humans are able to rapidly understand scenes by utilizing concepts extracted from prior experience. Such concepts are diverse, and include global scene descriptors, such as the weather or lighting, as well as local scene descriptors, such as the color or size of a particular object. So far, unsupervised discovery of concepts has focused on either modeling the global scene-level or the local object-level factors of variation, but not both. In this work, we propose COMET, which discovers and represents concepts as separate energy functions, enabling us to represent both global concepts as well as objects under a unified framework. COMET discovers energy functions through recomposing the input image, which we find captures independent factors without additional supervision. Sample generation in COMET is formulated as an optimization process on underlying energy functions, enabling us to generate images with permuted and composed concepts. Finally, discovered visual concepts in COMET generalize well, enabling us to compose concepts between separate modalities of images as well as with other concepts discovered by a separate instance of COMET trained on a different dataset[*].

## 1 Introduction

Human intelligence is characterized by its ability to learn new concepts, such as the manipulation of a new tool from only a few demonstrations [1]. Essential to this capability is the composition and re-utilization of previously learned concepts to accomplish the task at hand [36]. This is especially apparent in natural language, which is often described as a tool for making 'infinite use of finite means' [8]. Previously acquired words can be infinitely nested using a set of grammatical rules to communicate an arbitrary thought, opinion, or state one is in. In this work, we are interested in constructing a system that can discover, in an unsupervised manner, a broad set of these compositional components, as well as subsequently combine them across distinct modalities and datasets.

For obtaining such decompositions, two separate lines of work exist. The first focuses on obtaining global, holistic, compositional factors by situating data points, such as human faces, in an underlying (fixed) factored vector space [24, 48]. Individual factors, such as emotion or hair color, are represented as independent dimensions of the vector space, with recombination between factors corresponding to the recombination of the underlying dimensions. Due to the fixed dimensionality of the vector space, multiple instances of a single factor, such as lighting, may not be combined, nor can individual

---

[*]Code and data available at https://energy-based-model.github.io/comet/

35th Conference on Neural Information Processing Systems (NeurIPS 2021).

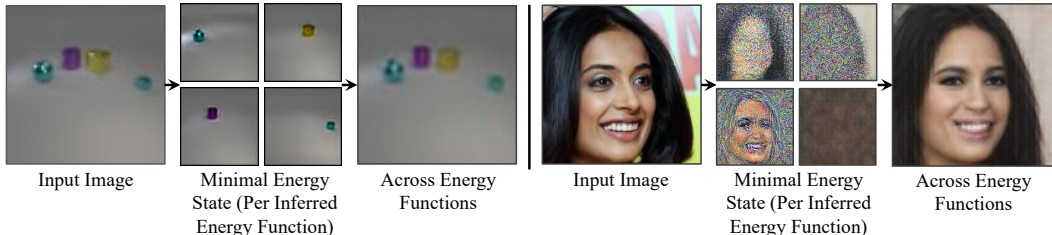

Input Image  Minimal Energy  Across Energy  Input Image  Minimal Energy  Across Energy
State (Per Inferred  Functions  State (Per Inferred  Functions
Energy Function)  Energy Function)

Figure 1: COMET decomposes images into a set of energy functions. The minimal energy state across all energy functions reconstructs the input image. The minimal energy states of individual energy functions capture particular aspects of an image, in the form of local factors of variations such as individual objects, or global factors of variation such as hair color, background lighting, facial expression, or skin color. **Note that reconstructed face images in the remainder of the paper are synthetic and not existing human faces.**

factored vector spaces from separate datasets be combined, i.e. the facial expression in an image from one dataset, and the background lighting in an image from another.

To address this weakness, a separate line of work decomposes a scene into a set of underlying 'object' factors. Each object factor represents a separate set of pixels in an image, as defined by a disjoint segmentation mask [22, 47]. Such a representation allows for the composition of individual factors by compositing the segmentation masks. However, by explicitly constraining the decomposition to be in terms of disjoint segmentation masks, relationships between individual factors of variation become more difficult to represent and capture global factors describing a scene.

In this work, we propose instead to decompose a scene into a set of factors represented as *energy functions*. An individual energy function represents a factor by assigning low energy to scenes with said factor and high energy to scenes where said factor is absent. A scene is then generated by optimizing the sum of energies for all factors. Multiple factors can be composed together, by summing the energies for each individual factor. Simultaneously, these individual factors are defined across the entire scene, allowing energy functions to represent global factors (lighting, camera viewpoint) and local factors (object existence).

Our work is inspired by recent work [12] which shows that energy based models may be utilized to represent flexible compositions of both global and local factors. However, while [12] required supervision, as labels were used to represent concepts, we aim to decompose and discover concepts in an unsupervised manner. In our approach, we discover energy functions from separate data points by enforcing compositions of these energy functions to recompose data.

Our work is further inspired by the ability of humans to effortlessly and reliably combine concepts gathered from disparate experiences. As a separate benefit of our approach, we show that we may take components inferred by one instance of our model trained on one dataset, and compose it with other components inferred by other instances of our model trained on separate datasets.

We provide analysis showing why our approach, COMET[†], is favorable compared to existing unsupervised approaches for decomposing scenes. We contribute the following: First, we show COMET provides a unified framework enabling us to decompose images into both global factors of variation as well as local factors of variation. Second, we show that COMET enables us to scale to more realistic datasets than previous work. Finally, we show that components obtained by COMET generalize well, and are amenable to compositions across different modes of data, and with components discovered by other instances of COMET.

## 2   Related Work

Our work is related to research in the areas of global factor disentanglement [4, 7, 23, 34, 38, 43] and independent component analysis (ICA) [2, 9, 25–27, 31, 32, 42, 51]. Work in said areas focus on discovering an underlying global latent space which describes the input space. In contrast, we decompose data into a set of compositional vector spaces. This enables our approach to compose multiple instances of one factor together, as well as compose factors across distinct datasets.

---

[†]short for COMposable Energy neTwork

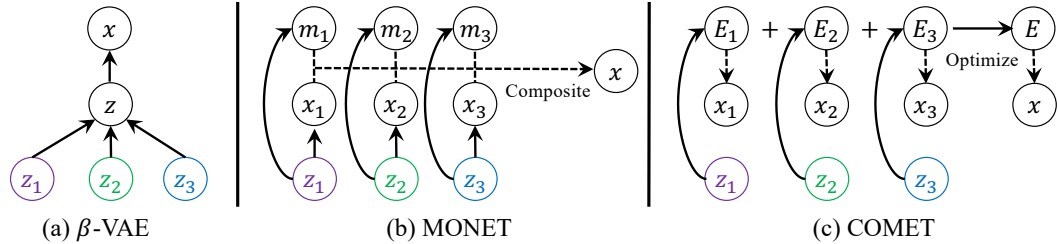

| (a) $\beta$-VAE | (b) MONET | (c) COMET |

Figure 2: Illustration of distinct approaches to composing components $z_1, z_2, z_3$ into an image $x$. (a) $\beta-$VAE utilizes a global decoder to map components to images. (b) MONET composits disjoint segmentation masks representing each image. (c) COMET defines an energy function per component, and optimizes the sum over the set of energy functions for each component.

Our work is also related to a large body of existing work on unsupervised object discovery [5, 10, 14–17, 21, 22, 35, 39, 45, 46, 49]. Such methods seek to decompose the scene into its underlying compositional objects by independently segmenting out each individual object in the scene. In contrast, our approach represents individual objects without the need of an explicit segmentation mask, enabling our approach to represent global relations between objects.

Our work draws on recent work in energy based models (EBMs) [11, 13, 18, 20, 33, 41, 44, 50]. Our underlying energy optimization procedure to generate samples is reminiscent of Langevin sampling, which is used to sample from EBMs [11, 41, 50]. Most similar to our work is that of [12], which proposes composing EBMs for compositional visual generation. Different from [12], we study how we may discover factors in an unsupervised manner from data.

## 3 Composable Energy Networks

Let $\mathcal{D} = \{X\}$ be a set of images $x \in \mathbb{R}^D$. Our goal is to obtain a set $Z$ of components $\{z_0, z_1, \ldots, z_k\}$ for each image $x$ representing the underlying factors of variation in the image, where each component $z \in \mathbb{R}^M$. We first discuss how we represent $Z$ as a set of energy functions in Section 3.1. We then discuss how COMET learns to decompose data in an unsupervised manner into a set of components in Section 3.2.

### 3.1 Composing Factors of Variation as Energy Functions

In prior work for decomposing images into a set of components $Z$, approaches such as $\beta$-VAE [23] utilize a parameterized, non-adaptive, feedforward decoder to map the set of components to an image. Due to this, the method is unable to compose multiple instances of the same component or a larger number of components then seen during training. On the other hand, approaches such as MONET [5] utilize a decoder with shared weights to process each individual component. Unfortunately, the components are decoded independently, preventing relationships between individual components from being captured.

To flexibly and generically compose a set of components $Z$, we require an approach that (1) utilizes a decoder that is shared across each component, such that variable-sized sets of components may be encoded, while also (2) decodes components jointly such that the relationships between individual components are covered. To construct an approach with the aforementioned desiderata, we propose to represent each component $z$ as an energy function. We illustrate our approach and its differences from prior work in Figure 2.

**Representing Components as Energy Functions.** Given a single component $z$, we encode the factor using an energy function $E_\theta(x, z) : \mathbb{R}^D \times \mathbb{R}^M \to \mathbb{R}$, which is learned to assign low energy to all images $x$ which contain the component and high energy to all other images. To obtain an image $x'$ with a component $z$, we solve for the expression $x' = \arg\min_x E_\theta(x; z_i)$. Note that in contrast to prior work composing energy functions [12], our energy functions have no probabilistic interpretation.

**Composing Energy Functions.** Given multiple components, we represent a set of components $\{z_i\}$ by performing a summation $\sum_i E_\theta(x, z_i)$ over each component's energy function. To obtain an image $x'$ for the set of components $Z$, we solve for the expression $x' = \arg\min_x \sum_i E_\theta(x; z_i)$.

While both COMET and MONET rely on summation as a tool for composing components, MONET sums components in the image domain, while COMET sums the cost functions representing each component together. By summing cost functions, each component may combine with other components in a non-linear manner, enabling us to model more complex relationships between individual components.

As a result, our generation $x'$ is the byproduct of jointly minimizing each individual energy function $E_\theta(x, z_i)$, and thus our generation contains each component. In addition, each individual component $z_i$ is parameterized by the same energy function $E_\theta$, enabling us to model a different number of components.

## 3.2 Unsupervised Decomposition of Composable Energy Functions

We next discuss how COMET discovers a set of composable energy functions from an input image $x_i$. In Section 3.1, we discuss that a set of components $Z = \{z_i\}$ may be encoded as a set of composable energy functions using the expression $\arg\min_x \sum_i E_\theta(x; z_i)$. To discover the set of components $Z$ in an unsupervised manner, we train by recomposing the input image $x_i$

$$\mathcal{L}_{\text{MSE}}(\theta) = \|\arg\min_x (\sum_k E_\theta(x; \text{Enc}_\theta(x_i)[k])) - x_i\|^2. \tag{1}$$

We utilize a learned neural network encoder $\text{Enc}_\theta(x_i)$ to infer a set of components $z_k$. In practice, evaluating the expression $\arg\min_x \sum_k E_\theta(x; \text{Enc}_\theta(x_i)[k])$, is computationally intractable. We thus approximate the $\arg\min$ operation with respect to $x$ via $N$ steps of gradient optimization, with an approximate optimum $x_i^N$ obtained as

$$x_i^N = x_i^{N-1} - \lambda \nabla_x \sum_k E_\theta(x_i^{N-1}; \text{Enc}_\theta(x_i)[k]). \tag{2}$$

In the above expression, we initialize optimization of $x_i^0$ with uniform noise with $\lambda$ representing the step size for each gradient step. Sample $x_i^n$ denotes the result after $n$ steps of gradient descent. We train our energy function $E_\theta$ using the modified objective $\mathcal{L}_{\text{MSE}} = \sum_{n=1}^N \|x_i^n - x_i\|^2$, where we minimizing MSE w.r.t $x_i^n$. To train parameters of $E_\theta$ we use automatic differentiation to compute gradients with respect to each optimization step depicted in Equation 2, where for training stability, we truncate gradient backpropogation to the previous time step.

We provide pseudocode for training our model in Algorithm 1. While the approach is simple, we find that it performs favorably in both global disentanglement as well as object-level disentanglement, and requires no additional objectives or priors to shape underlying latents. We utilize the same energy architecture $E_\theta$ throughout all experiments, and present details in the appendix.

**Algorithm 1** Training algorithm for COMET.
___
**Input:** data dist. $p_D(x)$, step size $\lambda$, number of gradient steps $N$, encoder $\text{Enc}_\theta$, energy function $E_\theta$, energy components K
**while** not converged **do**
    $x_i \sim p_D$
    ▷ *Encode components $z_i^k$ from $x_i$*
    $z_i^1, \ldots, z_i^K \leftarrow \text{Enc}_\theta(x_i)$
    ▷ *Optimize sample $x_i^0$ via gradient descent:*
    $x_i^0 \sim \mathcal{U}(0,1)$
    **for** gradient step $n = 1$ to $N$ **do**
        $x_i^n \leftarrow x_i^{n-1} - \lambda \nabla_x \sum_{k=1}^K E_\theta(x_i^{n-1}; z_i^k)$
    **end for**
    ▷ *Optimize objective $\mathcal{L}_{MSE}$ wrt $\theta$:*
    $\Delta\theta \leftarrow \nabla_\theta \sum_{n=1}^N \|x_i^n - x_i\|^2$
    Update $\theta$ based on $\Delta\theta$ using optimizer
**end while**
___

**Controlling Local and Global Decompositions.** In many scenes, both global and local factor decompositions are valid. To control the decomposition obtained by COMET, we bias the system towards inferring local factor decompositions by utilizing i) low latent dimensionality and ii) positional embeddings [37], both of which have been used in previous object discovery works [5]. This bias serves to encourage and enable models to focus on local patches of an image. We provide analysis of the effect of each inductive bias on the decomposition in Section 5.2.

## 4 Complexity Analysis of Energy Function Compositions

In this section, we provide complexity-theoretic motivation for composing functions together using energy functions. Let $Z$ denote a set of components, with individual components $z_1, \ldots, z_K \in \mathbb{R}^M$.

Let $\mathcal{F}$ be the space of functions $f: Z \to \mathbb{R}^D$ for mapping sets of components to an output vector $\boldsymbol{x}$. We consider two subspaces for $\mathcal{F}$.

**Composition of Energy Functions.** Let $\mathcal{F}_{\text{energy}} \subseteq \mathcal{F}$ be the subspace of functions of the form $\arg\min_{\boldsymbol{x}} \sum_{1 \le k \le K} E(\boldsymbol{x}, \boldsymbol{z}_k)$, where each $E(\boldsymbol{x}, \boldsymbol{z}_m)$ is a function from $\mathbb{R}^D \times \mathbb{R}^M \to \mathbb{R}$. This subspace corresponds to the set of compositions realized by COMET.

**Composition of Segmentation Masks.** We next consider the subspace of functions $\mathcal{F}_{\text{mask}} \subseteq \mathcal{F}$ consisting of functions $f$ of the form $\sum_{1 \le k \le K} m_k(\boldsymbol{z}_k) f_k(\boldsymbol{z}_k)$ where $f(\boldsymbol{z}) : \mathbb{R}^M \to \mathbb{R}^D$ and $m_k(\boldsymbol{z}_k) : \mathbb{R}^M \to \{0, 1\}^D$. Here, $m_k(\boldsymbol{z}_k)$ represents a segmentation mask in $\mathbb{R}^D$. This composition is used in object decomposition methods, such as [5, 22].

We first show that compositions of energy functions are more expressive than compositions of segmentation masks.

**Remark 1.** *The subspace $\mathcal{F}_{energy}$ is a strict superset of the subspace $\mathcal{F}_{mask}$.*

*Proof.* Any function of the form $\sum_{\boldsymbol{z} \in Z} m_k(\boldsymbol{z}) f(\boldsymbol{z})$ is equivalent to $\arg\min_{\boldsymbol{x}} \sum_{\boldsymbol{z} \in Z} E'(\boldsymbol{x}, \boldsymbol{z})$, where $E'(\boldsymbol{x}, \boldsymbol{z}) = m_k(\boldsymbol{z})(\boldsymbol{x} - f(\boldsymbol{z}))^2$, so $\mathcal{F}_{\text{mask}} \subseteq \mathcal{F}_{\text{energy}}$. In the other direction, note that according to the definition of $\mathcal{F}_{\text{mask}}$, the presence of a particular component $\boldsymbol{z}_i$ assigns a fixed value to the output for the nonzero entries of $m_k(\boldsymbol{z}_k) f(\boldsymbol{z}_k)$, irrespective of the value of the other components. In contrast, the optimal value of $\boldsymbol{x}$ in $\mathcal{F}_{\text{energy}}$ depends on the value of all components. A constructive example of this is the set of energy functions over one-dimensional $\boldsymbol{x}$ where $E(x, \boldsymbol{z}_1) = (x - 2)^2$, $E(x, \boldsymbol{z}_2) = (x - 3)^2$, $E(x, \boldsymbol{z}_3) = (x - 4)^2$. Thus, we have that $\arg\min_x E(x, \boldsymbol{z}_1) + E(x, \boldsymbol{z}_2) \ne \arg\min_x E(x, \boldsymbol{z}_1) + E(x, \boldsymbol{z}_3)$. Therefore, there are functions in $\mathcal{F}_{\text{energy}}$ that are not in $\mathcal{F}_{\text{mask}}$. $\square$

Next, we show that even in the setting in which we represent a single component $\boldsymbol{z}$, learning a function to approximate $E(\boldsymbol{x}, \boldsymbol{z})$ is more computationally efficient than learning a function $f(\boldsymbol{z})$ that approximates $\arg\min_{\boldsymbol{x}} E(\boldsymbol{x}, \boldsymbol{z})$. Intuitively, an energy function $E(\boldsymbol{x}, \boldsymbol{z})$ can be seen as a verifier of a set of constraints, with the value of $E(\boldsymbol{x}, \boldsymbol{z})$ being low when all constraints are satisfied. Approximating the function $\arg\min_{\boldsymbol{x}} E(\boldsymbol{x}, \boldsymbol{z})$ corresponds to generating a solution given a set of constraints, which is well-known in complexity theory to be much harder than verifying the constraints. Thus, to enable formal analysis of $E(\boldsymbol{x}, \boldsymbol{z})$, we reduce the 3-SAT [28] problem to an energy function $E(\boldsymbol{x}, \boldsymbol{z})$.

Given a 3-SAT formula $\phi$ with $D$ variables and $K$ clauses, we encode $\phi$ using an energy function $E(\boldsymbol{x}, \boldsymbol{z}) := \sum_{1 \le k \le K} e_k(\boldsymbol{x}, \boldsymbol{z}[k])$, where, $\boldsymbol{x}$ encodes an assignment to all the variables, $\boldsymbol{z}[k]$ represents the dimensions of $\boldsymbol{z}$ encoding the $k^{\text{th}}$ clause, and each $e_k$ is a function which has energy 0 if the assignment $\boldsymbol{x}$ satisfies the $k^{\text{th}}$ clause, and has energy 1 otherwise. To encode a clause using $\boldsymbol{z}[k]$, we utilize an ordinal representation (e.g. $\boldsymbol{z} = [1, 2, 3]$ to represent the clause $(x_1 \wedge x_2 \wedge x_3)$), and round non-integer coordinates of $\boldsymbol{x}$ and $\boldsymbol{z}$ to the nearest integer. We assume the Exponential Time Hypothesis (ETH)[28], which states that checking the satisfiability of a 3-SAT formula takes time exponential in the sum of the number of variables and the number of clauses.

**Remark 2.** *There exists an energy function $E(\boldsymbol{x}, \boldsymbol{z}_1)$ which can be evaluated at any input in time polynomial in the number of dimensions of $\boldsymbol{x}$ but for which the computational complexity of evaluating $f(\boldsymbol{z}) := \arg\min_{\boldsymbol{x}} E(\boldsymbol{x}, \boldsymbol{z}_1)$ is (worst-case) exponential in the number of dimensions of $\boldsymbol{x}$.*

*Proof.* If we utilize the 3-SAT energy function $E(\boldsymbol{x}, \boldsymbol{z})$ defined above. ETH implies that solving the 3-SAT problem, corresponding to computing $f(\boldsymbol{z}) = \arg\min_{\boldsymbol{x}} E(\boldsymbol{x}, \boldsymbol{z}_1)$, is exponential in dimension of $\boldsymbol{x}$. In contrast, evaluating each entry of our defined $E(\boldsymbol{x}, \boldsymbol{z}_1)$ is polynomial in dimension of $\boldsymbol{x}$. $\square$

Our remark shows that it is computationally advantageous to learn an energy function $E(\boldsymbol{x}, \boldsymbol{z})$ as opposed to a decoder $f(\boldsymbol{z})$. To realize the exponential number of computations needed to compute $f(\boldsymbol{z})$, significantly more capacity is necessary to represent $f(\boldsymbol{z})$ in comparision to $E(\boldsymbol{x}, \boldsymbol{z})$. We further show that as we compose multiple 3-SAT energy functions together, learning a decoder $f(\boldsymbol{z}_1, \ldots, \boldsymbol{z}_k) := \arg\min_{\boldsymbol{x}} \sum_k E(\boldsymbol{x}, \boldsymbol{z}_k)$ that represents the optimization process scales exponentially with the number of energy components.

**Remark 3.** *There exists a composition of energy functions $\sum_k E(\boldsymbol{x}, \boldsymbol{z}_k)$ which can be evaluated in time polynomial in the number of components $k$, but for which the computational complexity for evaluating $f(\boldsymbol{z}_1, \ldots, \boldsymbol{z}_k) := \arg\min_{\boldsymbol{x}} \sum_k E(\boldsymbol{x}, \boldsymbol{z}_k)$ is (worst-case) exponential in the components of components $k$.*

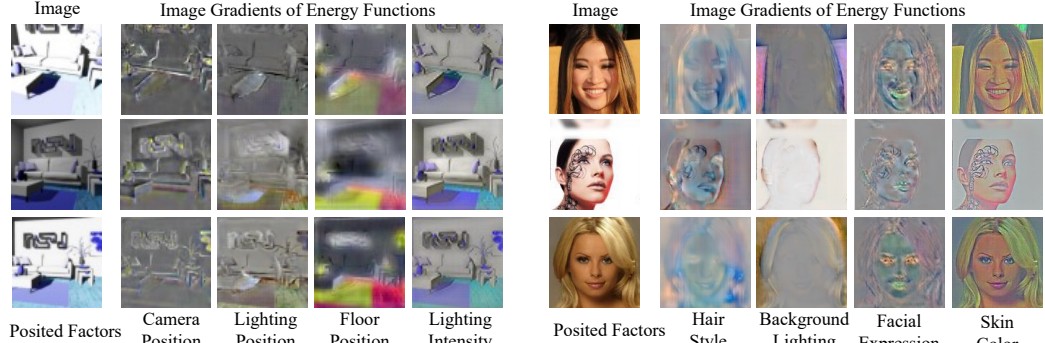

Figure 3: **Global Decomposition.** Illustration of energy functions gradients of each decomposed energy function in COMET on Falcor3D (**left**) and CelebA-HQ (**right**) datasets. Gradients correspond to aspects of images each energy function pays attention to. Discovered energy functions are labeled with the posited factors they capture, as determined by qualitative inspection.

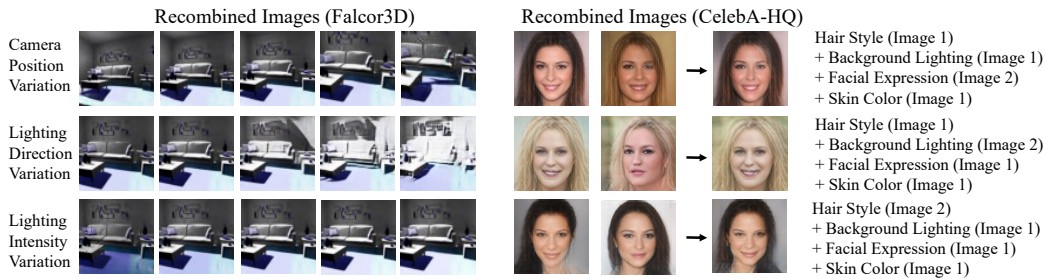

Figure 4: **Global Factor Recombination.** Illustration of recombination of energy functions on Falcor3D and CelebA-HQ datasets. In Falcor3D (**left**), we illustrate variation of a single energy function, which elicits changes in camera position, lighting direction and lighting intensity variation. In CelebA-HQ (**right**), we recombine discovered energy functions across two separate images (energy functions labeled through qualitative visualization in Figure 3).

*Proof.* Given $k$ separate 3-SAT energy functions $E(\boldsymbol{x}, \boldsymbol{z}_k)$, minimizing the composed energy function, $f(\boldsymbol{z}_1, \ldots, \boldsymbol{z}_k) = \arg\min_{\boldsymbol{x}} \sum_k E(\boldsymbol{x}, \boldsymbol{z}_k)$ corresponds to solving all 3-SAT encoded clauses across $k$ energy functions. ETH implies the computational complexity of evaluating $f(\boldsymbol{z}_1, \ldots, \boldsymbol{z}_k)$ is exponential in $k$ while the evaluation of $k$ energy functions is polynomial in $k$. □

Here as well, to represent the exponential number of computations, significantly more capacity is necessary to realize the computation $f(\boldsymbol{z}_1, \ldots, \boldsymbol{z}_k)$, which scales with the number of components $k$, in comparison to $\sum_k E(\boldsymbol{x}, \boldsymbol{z}_k)$. Thus, remark 2 and 3 show that it can be efficient, from a learning perspective, to decode individual datapoints with energy functions.

## 5 Evaluation

We quantitatively and qualitatively show that COMET can recover the global factors of variation in an image in Section 5.1, as well as the local factors in an image in Section 5.2. Furthermore, we show that the components captured by COMET can generalize well, across separate modalities in Section 5.3.

### 5.1 Global Factor Disentanglement

We assess the ability of COMET to decompose global factors of variation in scenes consisting of lighting and camera illumination from Falcor3D (NVIDIA high-resolution disentanglement dataset) [40], scene factors of variation in CLEVR [29], and face attributes in real images from CelebA-HQ [30]. For all experiments, we utilize the same convolutional encoder to extract sets of latents from each dataset, and utilize a latent dimension of 64 for each separately inferred latent. We provide additional training algorithm and model architecture details in the appendix.

**Decomposition and Reconstructions.** In Figure 3, on Falcor3D and CelebA-HQ, we illustrate the underlying image gradients of each decomposed energy function. Such gradients correspond to

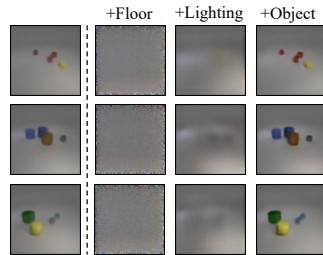

|  | +Floor | +Lighting | +Object |
|---|---|---|---|

Input  Generation (Per Additive Component)

Figure 5: **CLEVR Decomposition.** Generations on CLEVR when optimizing over an increasing number of energy functions.

| Model | Dim ($D$) | $\beta$ | Decoder Dist. | BetaVAE | MIG | MCC |
|---|---|---|---|---|---|---|
| COMET* | 64 | – | – | $99.41 \pm 0.15$ | $19.63 \pm 2.49$ | $76.55 \pm 1.35$ |
| $\beta$-VAE | 64 | 4 | Gaussian | $83.57 \pm 8.05$ | $10.90 \pm 3.80$ | $66.08 \pm 2.00$ |
| $\beta$-VAE | 32 | 4 | Gaussian | $79.77 \pm 10.95$ | $7.14 \pm 5.44$ | $57.48 \pm 6.04$ |
| $\beta$-VAE | 256 | 4 | Gaussian | $80.76 \pm 4.55$ | $10.94 \pm 0.58$ | $66.14 \pm 1.81$ |
| $\beta$-VAE | 64 | 16 | Gaussian | $74.71 \pm 1.57$ | $9.33 \pm 3.72$ | $57.28 \pm 2.37$ |
| $\beta$-VAE | 64 | 1 | Gaussian | $81.61 \pm 6.75$ | $6.51 \pm 3.38$ | $58.73 \pm 6.31$ |
| $\beta$-VAE | 64 | 4 | Bernoulli | $84.23 \pm 3.51$ | $8.96 \pm 3.53$ | $61.57 \pm 4.09$ |
| InfoGAN | 64 | – | – | $79.65 \pm 1.69$ | $2.48 \pm 1.11$ | $52.67 \pm 1.91$ |
| MONet* | 64 | – | – | $93.13 \pm 1.02$ | $13.94 \pm 2.09$ | $65.72 \pm 0.89$ |

Table 1: **Disentanglement Evaluation.** Mean and standard deviation (s.d.) metric scores across 3 random seeds on the Falcor3D dataset. COMET enables better disentanglement according to 3 common disentanglement metrics across different runs and seeds for training $\beta$-VAE, InfoGAN and MONet baselines. Note that * denotes that PCA was used as a postprocessing step.

aspects of the input image each energy function pays attention. Through qualitative examination, we posit that the individual inferred energy functions for Falcor3D correspond to camera position, lighting direction, floor position and lighting intensity (shown from left to right in Figure 3). Such a correspondence can be seen for example by the fact that the camera position energy function exhibits sharp gradients with respect to edges in an image, while the lighting direction energy function exhibits sharp gradients with respect to the underlying shadows in an image. In a similar manner, we hypothesize that the individual inferred energy functions for CelebA-HQ correspond to hair color, background lighting, facial expression and skin color.

In Figure 5, we show energy function decompositions of CLEVR, where we illustrate generations when composing an increasing number of inferred energy functions. By adding individual energy functions, we find that the generation transitions from consisting only of objects, to consisting of objects & floor to consisting of objects, floor, & lighting.

**Recombination.** To further verify that each individual energy function captures the expected decomposition of factors described earlier, we recombine individual energy functions representing each component in the Falcor3D and CelebA-HQ datasets in Figure 4. In the left side of Figure 4, we vary an energy function representing a single factor, while keeping the remaining factors fixed. By varying energy functions in such a manner, we are able to capture camera position, lighting direction and lighting intensity variation (with camera position variation captured by varying energy functions for both floor position and camera position). In the right side of Figure 4, we can recombine discovered energy functions representing facial expression, hair color, and background lighting of one image with that of another. We find that by recombining individual energy functions representing each individual factor, we are able to reliably swap factors across images.

**Quantitative Comparison.** Finally, we evaluate the learned representations on disentanglement. In Falcor3D [40], each image corresponds to a combination of 7 factors of variation; lighting intensity, lighting $x$, $y$ & $z$ direction, and camera $x$, $y$ & $z$ position. We consider three commonly used metrics for evaluation, the BetaVAE metric [23], the Mutual Information Gap (MIG) [7], and the Mean Correlation Coefficient (MCC) [25]. See Appendix C of [34] for extended descriptions of metrics.

Standardized metrics in disentanglement assume flattened model latents, i.e. relate the 7 factors of variation to the $D$-dimensional encoding of the corresponding image. However, as discussed, our method extracts sets of latents from images. We thus extract $D$ principal components from the set of latents, which we then compare with $\beta$-VAE. In Table 1, we find that our approach performs better than that of $\beta$-VAE across hyperparameter settings, as well as additional baselines of MONet and InfoGAN.

## 5.2 Object Level Decomposition

Next, we assess the ability of COMET to decompose object-level factors of variation in an image. We evaluate the ability of COMET to isolate individual tetris blocks in the Tetris dataset from [22], and individual object segmentations in the CLEVR dataset [29]. To bias energy functions to local object-level variations, we utilize small latent dimension (16), and add positional embeddings to images. To extract repeated object-level structure in an image, we utilize a recurrent network as our

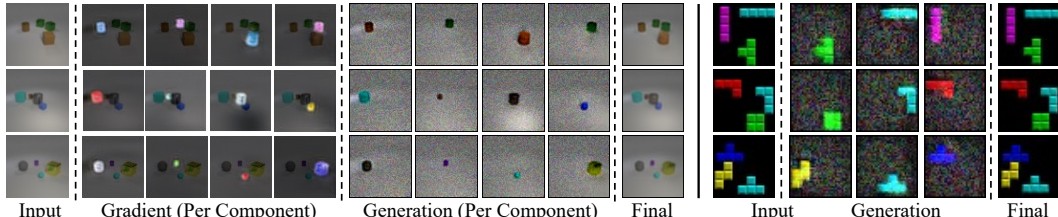

| Input | Gradient (Per Component) | Generation (Per Component) | Final | Input | Generation | Final |

Figure 6: **Object Decomposition.** COMET can decompose underlying object-level factors of variation in CLEVR (left) and Tetris (right). Individual energy functions exhibit sharp gradients with respect to each object in CLEVR.

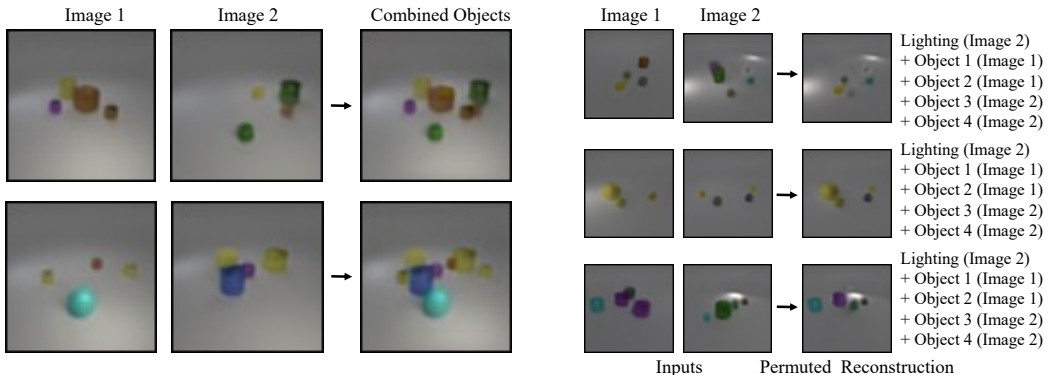

Figure 7: **Compositional Generalization (Left).** We may combine energy functions from COMET on two separate images together to generate a new image with 8 objects. **Global/Local Factor Recombination (Right).** COMET is able to discover energy functions corresponding to global and local factors of variation simultaneously from a given image. We can recombine individual energy functions corresponding to global factors of variation (lighting) and local factors of variation (cube position).

encoder, similar to prior unsupervised object discovery work [5]. We provide additional training algorithm and model architecture details for each experimental setup in the appendix.

**Decomposition and Reconstructions.** We illustrate object-level decomposition of individual energy functions on CLEVR and Tetris in Figure 6. In both settings, we find that individual energy functions correspond to the underlying objects in the scene. On CLEVR, while in Figure 5 we obtain global factor decompositions, by biasing our energy function to object level decompositions we obtain individual cube decompositions in Figure 6.

**Combinatorical Recombination.** We next consider recombining inferred object components. In Figure 7 (left), we combine energy functions for individual CLEVR objects across two separate scenes. We find that by combining 8 energy functions corresponding to 4 objects from separate images, we are able to successfully generate an image containing all 8 objects, and with some consistency with respect to occlusion. In contrast, objects composed together through MONET do not respect occlusion, as they are represented as disjoint segmentation masks.

**Quantitative Comparisons.** For quantitative comparison with MONET, we create approximate segmentation masks per energy function in CLEVR by thresholding the gradients (illustrated in Figure 6) of the energy function. We find that our approach obtains an ARI [22] of 0.916 and a mean segmentation covering of 0.713. In contrast, we find MONET obtains an ARI of 0.873 and a mean segmentation covering of 0.701.

**Decompositions of Global and Object Level Factors.** COMET is distinct from previous approaches for unsupervised decomposition in that it is able to decompose both global and local factors of variation. We find that we are further able to *control* the capture of either a particular global or local factor. In particular, we find that by conditioning a particular energy function with a positional embedding added to the image [37] and a small latent dimension we may effectively bias a energy function to capture an object factor of the scene. In contrast, a larger latent size biases an energy function to capture a global factor of the scene. In Table 2, we investigate the extent of these two effects in enabling the capture of local factors in CLEVR as measured by ARI. We find that the

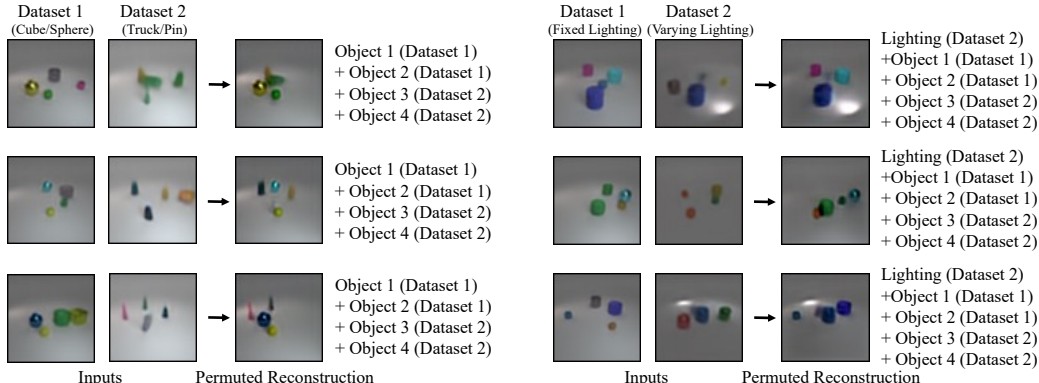

Figure 8: **Cross Dataset Composition.** Energy functions from COMET discovered in one dataset may be combined with energy functions from a separate instance of COMET discovered on a different dataset. In the **left panel**, we recombine energy functions representing cube/sphere objects discovered in CLEVR with energy functions representing truck/pin objects discovered in CLEVR Toy. We are able to generate novel images consisting of objects from both datasets. In the **right panel**, we recombine energy functions representing separate objects in CLEVR with energy functions representing both lighting and individual objects from CLEVR Lighting. We are able to generate novel images consisting of novel combinations of both objects and lighting.

addition of utilizing both small latent dimension and positional embedding enable more effective decomposition of underlying object factors.

By selectively enabling ourselves to specify both the underlying global and local factors of a scene, we may obtain an unsupervised decomposition of a scene with both local and global factors of variation. To test this, we render a novel dataset, *CLEVR Lighting*, by increasing the lighting variation in CLEVR (illustrated in Figure 7 (right)), and train COMET to infer 5 separate energy functions. We encourage the first energy function to capture a global factor of variation by removing the positional embedding from the first inferred energy function. In such a setting, we find COMET successfully infers lighting as the first energy function, and individual objects for the subsequent components, which we illustrate in the appendix. We

| Recurrent Encoder | Small Latent | Pos Embed | ARI ↑ |
|---|---|---|---|
| No | No | No | 0.413 |
| Yes | No | No | 0.407 |
| Yes | Yes | No | 0.641 |
| Yes | Yes | Yes | 0.889 |

Table 2: **Local Factor Inductive Bias.** Analysis of different inductive biases on underlying local factor decomposition as measured by ARI score on the CLEVR dataset.

present recombinations of these inferred components in Figure 7 (right). A limitation of our approach is that that lighting and object factors of variations are not completely disentangled. For example, in the the top row, the permuted reconstruction of the image has incorrect lighting in the center.

## 5.3 Compositional Factor Generalization

Finally, we assess the ability of components inferred from COMET to generalize. We study two separate settings of generalization. We first evaluate the ability of components to generalize to multi-modal inputs by training COMET to decompose images drawn from both CelebA-HQ and Danbooru datasets [3], as well as KITTI [19] and Virtual KITTI [6] datasets. We next assess the ability of components from COMET to compose with a separate instance of COMET. We train separate COMET models on CLEVR, CLEVR Lighting and an additional novel dataset, *CLEVR Toy*, which is rendered by replacing sphere/cylinder/cube objects with pin/boot/toy/truck objects.

**Cross Dataset Recombination.** COMET may compose inferred components from one dataset with components discovered by a separate COMET trained on a separate dataset. We validate this in Figure 8 (left), where we recombine components specifying objects in CLEVR and CLEVR Toy. In Figure 8 (right), we further recombine components specifying objects in CLEVR with components specifying objects & lighting in CLEVR Lighting.

**Cross Modality Decomposition and Recombination.** We assess COMET decompositions on multi-modal datasets of Danbooru/CelebA-HQ and KITTI/Virtual KITTI in Figure 9. We find consistent decomposition of components between separate modes. Furthermore, we find that these components may be recombined between the separate modalities in Figure 10.

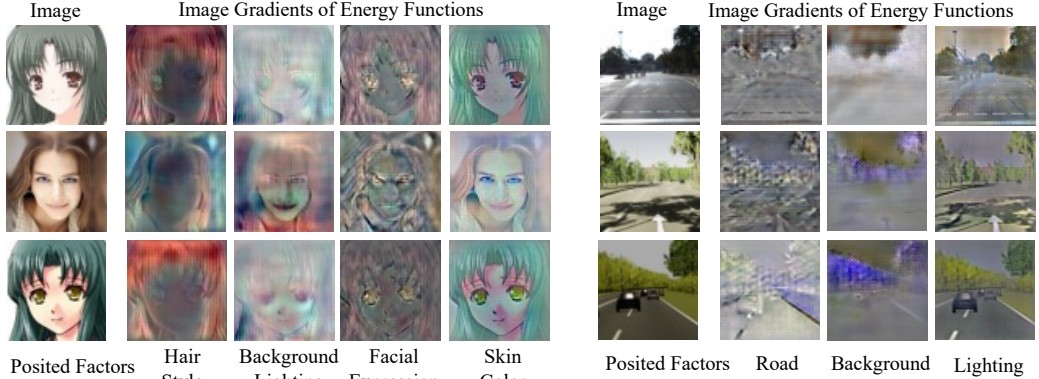

Figure 9: **Multi-modal Decomposition.** Illustration of energy gradients of each unsupervised decomposed energy function in COMET on sets of images drawn from distinct modalities. COMET discovers consistent decompositions between different modalities, where energy functions are labeled with the posited factor they capture (as determined from qualitative inspection). **(Left)** Visualization on images in the Danbooru and CelebA-HQ domains. D Energy functions are consistent with those discovered in Figure 3 (CelebA-HQ). **(Right)** Visualization on images in the KITTI and Virtual KITTI domains.

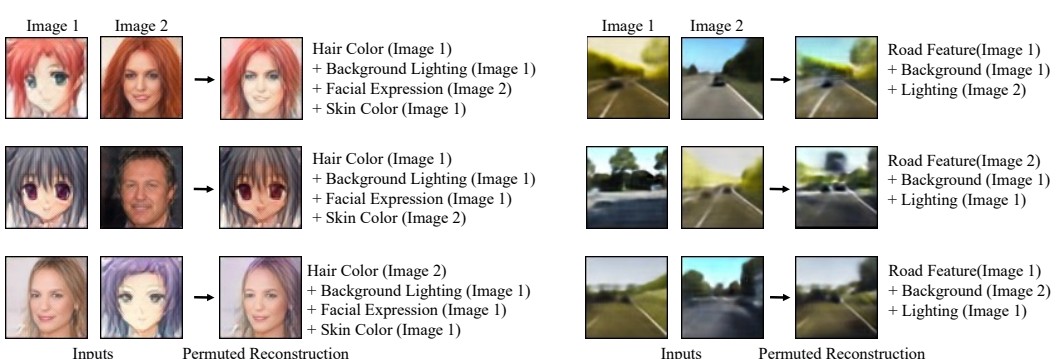

Figure 10: **Multi-modal Recombination.** Illustration of recombination of energy functions on multi-modal datasets of images. On images consisting of Danbooru and CelebA-HQ faces **(left)**, by recombining discovered energy functions, we are able to selectively change underlying facial expression, skin color and hair color across images from different domains. On images consisting of KITTI and Virtual KITTI images **(right)**, by recombining discovered energy functions, we are able to selectively change the underlying lighting, road and background between images of separate domains.

## 6 Conclusion

We have demonstrated an approach towards unsupervised learning of energy functions from images. We show how these functions encode both global and local factors of variations, and how they allow for further composition across separate modalities and datasets. A limitation of our current approach is that while it can compose factors across datasets that are substantially similar, compositions across datasets such as Falcor3D and CelebA-HQ are less interpretable. We posit that this is due to a lack of diversity in the underlying training data set. An interesting direction for future work would be to to train COMET on complex, higher diversity real world datasets, and observe subsequent recombinations of energy functions. We note that, as a consequence of using deep nets, our system is susceptible to dataset bias. Care must be put in ensuring a balanced and fair dataset if COMET is deployed in practice, as it should not serve to, even inadvertently, worsen societal prejudice.

**Acknowledgements**    We would like to thank Abhijit Mudigonda for giving helpful comments on the manuscript. Yilun Du is supported by a NSF graduate research fellowship. This work is in part supported by ONR MURI N00014-18-1-2846 and IBM Thomas J. Watson Research Center CW3031624. This work was supported by the German Federal Ministry of Education and Research (BMBF): Tübingen AI Center, FKZ: 01IS18039A. The authors thank the International Max Planck Research School for Intelligent Systems (IMPRS-IS) for supporting Yash Sharma.

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
