# A Unsupervised Learning of Compositional Energy Concepts Appendix

In this supplement, we provide additional empirical visualizations of our approach in Section A.1. Next, we provide details on experimental setup in Section A.2. The underlying code of the paper can be found at the paper website.

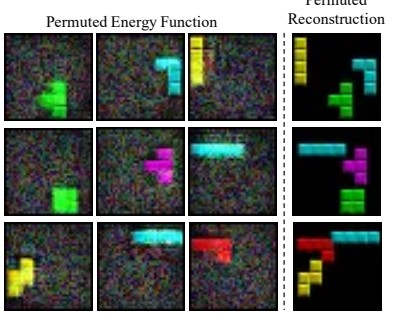

Figure A1: **Tetris Recombination.** Illustration of recombination of energy functions inferred by COMET on the Tetris dataset.

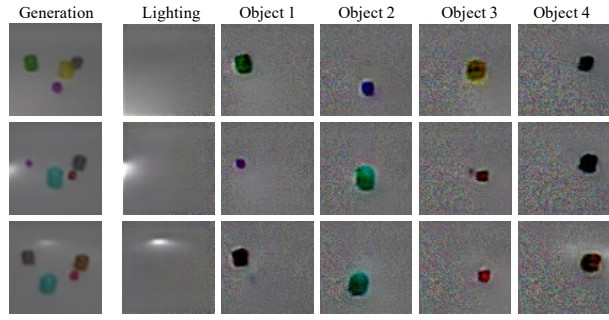

Figure A2: **Light Object Decomposition.** Illustration of decomposing CLEVR Lighting scenes into separate energy functions. COMET is able to decompose a scene into lighting and constituent objects.

| Model | Dim ($D$) | $\beta$ | Decoder Dist. | BetaVAE | MIG | MCC |
|---|---|---|---|---|---|---|
| $\beta$-VAE (Codebase 1) | 64 | 4 | Gaussian | $83.57 \pm 8.05$ | $10.90 \pm 3.80$ | $66.08 \pm 2.00$ |
| $\beta$-VAE (Codebase 2) | 64 | 4 | Gaussian | $79.99 \pm 9.65$ | $7.45 \pm 4.58$ | $61.03 \pm 5.49$ |
| $\beta$-VAE (Codebase 1) | 32 | 4 | Gaussian | $79.77 \pm 10.95$ | $7.14 \pm 5.44$ | $57.48 \pm 6.04$ |
| $\beta$-VAE (Codebase 2) | 32 | 4 | Gaussian | $88.01 \pm 8.18$ | $12.06 \pm 6.05$ | $63.42 \pm 5.94$ |
| $\beta$-VAE (Codebase 1) | 256 | 4 | Gaussian | $80.76 \pm 4.55$ | $10.94 \pm 0.58$ | $66.14 \pm 1.81$ |
| $\beta$-VAE (Codebase 2) | 256 | 4 | Gaussian | $83.93 \pm 6.42$ | $7.79 \pm 2.41$ | $61.89 \pm 2.78$ |
| $\beta$-VAE (Codebase 1) | 64 | 16 | Gaussian | $74.71 \pm 1.57$ | $9.33 \pm 3.72$ | $57.28 \pm 2.37$ |
| $\beta$-VAE (Codebase 2) | 64 | 16 | Gaussian | $71.30 \pm 4.24$ | $6.28 \pm 1.18$ | $55.14 \pm 3.10$ |
| $\beta$-VAE (Codebase 1) | 64 | 1 | Gaussian | $81.61 \pm 6.75$ | $6.51 \pm 3.38$ | $58.73 \pm 6.31$ |
| $\beta$-VAE (Codebase 2) | 64 | 1 | Gaussian | $86.42 \pm 3.33$ | $9.88 \pm 3.27$ | $62.79 \pm 3.21$ |
| $\beta$-VAE (Codebase 1) | 64 | 4 | Bernoulli | $84.23 \pm 3.51$ | $8.96 \pm 3.53$ | $61.57 \pm 4.09$ |
| $\beta$-VAE (Codebase 2) | 64 | 4 | Bernoulli | $93.86 \pm 1.74$ | $14.26 \pm 2.26$ | $68.34 \pm 1.94$ |

Table 1: **Disentanglement Evaluation.** Mean and standard deviation (s.d.) metric scores across 3 random seeds on the Falcor3D dataset. COMET enables better disentanglement across 3 common disentanglement metrics across different runs and seeds for training $\beta$-VAE. Note that $^*$ denotes that PCA was used as a postprocessing step.

## A.1 Additional Empirical Results

**Qualitative Result.** We provide additional empirical visualizations of the results presented in the main paper section 5.2. We illustrate the permuted energy functions in Tetris in Figure A1, and find that we are able to successfully permute different Tetris blocks across different images. We further show that our model is able to decompose an image into both objects and lighting factors in the CLEVR Lighting dataset in Figure A2.

**Quantitative Evaluation.** We provide an additional comparison of COMET to the $\beta$-VAE utilizing a separate codebase across 3 separate seeds in Table 1. We find that across different implementations COMET improves performance over the $\beta$-VAE.

## A.2 Model and Experimental Details

**Dataset Details.** For the CLEVR dataset, we utilize the dataset generation code from [2] to render images of scenes with 4 objects. For the CLEVR lighting dataset, we also utilize the code from [2] to

render the dataset but increase the lighting jitter to 10.0 for all settings. Finally, for the CLEVR toy dataset, we utilize the default CLEVR dataset generation code but replace the blend files of "sphere", "cylinder", and "cube" with that of "bowling pin", "boot", "toy" and "truck".

**Architecture Details.** In COMET we utilize a residual network to parameterize an underlying energy function. We illustrate the underlying architecture of the energy function in Figure 2. The energy function takes as input an image at $64 \times 64$ resolution and processes the image through a series of residual blocks combined with average pooling to obtain a final output energy. To condition the energy function on an input latent $z$, we linearly map $z$ to a separate per channel gain and bias in each residual block of the energy network, and use the resultant gain and bias vectors to modulate input features [5]. We remove normalization layers from our residual network.

To infer global factors from an input image, we utilize a convolutional encoder in Figure 3. The convolutional encoder maps an input image through a series of residual convolutional layers to obtain a set of distinct latent vectors. These resultant latent vectors are utilized to condition each separate energy function and correspond to individual global factors.

To infer local object factors in a scene, we utilize a convolutional encoder in combination with a recurrent network with spatial attention. We concatenate a positional embedding to the underlying input image of scene [4]. We then utilize a series of 3 residual layers to downsample input images at $64 \times 64$ resolution to a lower resolution $8 \times 8$ feature grid. We utilize a LSTM with an attention mechanism [1] to iteratively gather information from this feature grid to obtain a set of object latents representing the scene. We illustrate the overall architecture in Figure 4.

**Training Details.** Models for each dataset are trained for 12 hours on a single 32GB Volta machine. Models are trained utilizing the Adam optimizer [3] with a learning rate of 3e-4. We utilize 10 steps of optimization to approximate the minimal energy state of an energy function, with each gradient descent step utilizing a scalar multiplier of 1000. When training each model, the gradients are clipped to a magnitude of 1. Images are fit at $64 \times 64$ resolution, with a training batch size of 32. We utilize a latent dimension of 16 per component when extracting local factors of variation and a latent dimension of 64 when extracting global factors of variation. For global factor recombination, to obtain high-resolution results on real datasets, we utilize LPIPS loss as a replacement to MSE loss.

| |
| --- |
| 3x3 Conv2d, 64 |
| ResBlock Down 64 |
| ResBlock Down 64 |
| ResBlock Down 64 |
| Global Mean Pooling |
| Dense $\rightarrow$ 1 |

Table 2: Architecture of energy function.

| |
| --- |
| 3x3 Conv2d, 64 |
| ResBlock Down 64 |
| ResBlock Down 64 |
| ResBlock Down 64 |
| Global Mean Pooling |
| Dense $\rightarrow$ 64 |
| Dense $\rightarrow$ Latents |

Table 3: The model architecture used for the convolutional encoder.

| |
| --- |
| 3x3 Conv2d, 64 |
| ResBlock Down 64 |
| ResBlock Down 64 |
| ResBlock Down 64 |
| LSTM |
| Dense $\rightarrow$ 64 |
| Dense $\rightarrow$ Latents |

Table 4: The model architecture used for the recurrent encoder used in Section 5.2 of the main paper. We utilize a LSTM which operates on the spatial output of the residual network through attention [1].