# OpenReview forum: "Unsupervised Learning of Compositional Energy Concepts"
_NeurIPS.cc/2021/Conference — NeurIPS 2021 Poster_

### Official Review · Reviewer_2JFE · 2021-07-08

**Rating:** 6
**Confidence:** 4

**Summary:**

This paper proposes a novel method to composite the energies (statistical description of the images by the neural network). The author extracts the component energy by different latent codes for the image. The component latent code z controls the energy with the input image together in a conditional energy manner. The learned model can generate the image with the specified component codes inferred from the encoder network.

**Ethical Concerns:**

No.

**Limitations And Societal Impact:**

Yes.

**Main Review:**

1. Originality: The work is a novel combination of the well-know techniques, such as compositional energy and generation technique in “Compositional Visual Generation and Inference with Energy Based Models” and conditional generation technique in style transfer and the general conditioning layer.

2. Quality: The work is technically sound. I have several questions.

 (1). The author says at page2, line 74, “Our underlying energy optimization procedure is …Langevin sampling”, but in Eq. (2), the random term N(0, $\lambda$) of the Langevin dynamics is missed. Without the random diffusion, Eq. (2) degenerates to the gradient descent and may not guarantee the sampling x converging to the real distribution by the MCMC chain. Did author compare the performances of the inferring x between the Langevin dynamics and gradient descent?

  (2). The author claims to “capture independent factors without additional supervision” in the abstract, but from the main contribution of the paper, such as Eq. (1), (2) and Algorithm 1, I cannot find any guarantee for capturing the independent factors. The paper proposes method to compose the component energy, but does not design the method to learn the independent component factors from the encoder. Since all the component factors compete each other to explain the generated image, it is suggested to study the algorithm to guarantee each component factor is independent. As a consequence, the experimental results are less interpretable, that is, I need to guess the meaning of each component factor from the figures, such as Fig. 3, the experimental results about the interpretability can be further improved.

  (3). The quality of the generated images in all the experiments are not sharp and can be improved in a large part. It is suggested to use and generate high resolution images, which is a mature technology nowadays.

  (4). In the section 4 of Complexity Analysis, from the proof of remark 1, I can not find that the $\sum_z m_k(z)f(z)$ is equivalent to $\mathop{\arg\min}_x \sum_z m_k(z)(x-f(z))^2$, could you explain it?

3. Clarity: The paper is well written, but the results heavily depends on several techniques such as FILM, conditional EBMs, spatial attention, and the details are all missed. I know this is a conference paper and suffers space limitation, but the details is suggested to supply in the supplementary material. The current supplementary material still miss lots of details.

4. Significance: The result has referential significance. The other researchers could build on the idea to further develop other applications.



**Time Spent Reviewing:**

12

---

> ### Author Response · Authors · 2021-08-10
> **Reviewer 2JFE Response**
>
> Thank you for your detailed comments and feedback; we appreciate the time and attention you gave to reviewing our paper. We address your concerns for additional evaluation as well as textual clarifications below. Please let us know if you have any additional questions, we are happy to clarify or provide additional experiments.
>
> ------------------------
> **Q1) Utilization of Langevin Dynamics for Sampling.**
>  In the earlier stages of the project, we indeed explored utilizing Langevin sampling to optimize our underlying energy function as opposed to gradient descent. We did not observe much of a benefit towards adding noise to the gradient optimization procedure, and also note that our energy functions do not have a probabilistic interpretation from which we may run Langevin sampling. We will add this clarification to the related work, and will explicitly state that our learned energy functions have no probabilistic interpretation.
>
> **Q2) Independence of Latents in Energy Decomposition.**
>
> To more clearly showcase that underlying latents are independent in an energy decomposition, we show generations of COMET when varying only a single underlying energy function. In https://ibb.co/f8bbwhr, we illustrate on the left side, variations of a single energy function representing lighting intensity, lighting direction, and camera position, when combined with fixed remaining components. On the right side, we show ground truth renderings. We find that our renderings closely match ground truth renderings. We further quantitatively evaluate independence of latents through disentanglement, and provide additional evaluations across 3 seeds and additional baselines in https://ibb.co/QngJT61. We find that our model consistently outperforms alternatives at disentanglement, and thus exhibits independence of latents.
>
> We believe that an underlying reason that inferred latents are independent is due to the fact that each latent separately conditions on the same underlying energy network to reconstruct an image. Since the same energy network is utilized to reconstruct an input image, the most information-efficient manner to reconstruct the input is then to ensure each conditioned latent is mutually independent from other latents, i.e. there should be minimum redundancy. Our empirical results support this intuition, and find that t-SNEs of inferred latent vectors of separate components in global decomposition indeed cluster to separate components (https://ibb.co/mc6ZvZx). We will add a discussion of this in the method section of the paper.
>
>
> **Q3) High Resolution Image Generation.**
> See Section 1.3 of General Response.
>
> **Q4) Segmentation Mask and Energy Function Equivalence.**
> We describe why $\sum_k m_k(z)f(z)$ and $\arg⁡min_x ⁡\sum_k m_k(z)(x−f(z))^2$ are equivalent. First, note that the minimum of the individual energy function $\arg⁡min_x m_k(z)(x−f(z))^2$ corresponds to the expression $m_k(z)f(z)$, since then $(x-f(z))^2=0$ when $x=f(z)$. Next, note that by formulation, each of the individual $m_k(z)$ are mutually disjoint from each other, as they form a segmentation mask. Thus, the minimizer of the expression $\arg⁡min_x⁡ \sum_k m_k(z)(x−f(z))^2$ is $x=\sum_k m_k(z)f(z)$, as with such a value of $x$, the expression $\sum_k m_k(z)(x−f(z))^2=0$, the minimal value for a positive sum of squares. We will add this clarification to the revision.
>
> **Q5) Additional Clarification on FILM, conditional EBM, spatial attention.**
> Thank you for pointing out the lack of clarification of each module for us. This was an oversight in our original paper, and we will add additional paragraphs explaining each of these components in the appendix of the paper. Furthermore, we will add figures to help clarify each point. Below, we provide descriptions of each component (which we will further elaborate on in the appendix of the main paper) on each submodule of our approach.
>
> In COMET, we construct an energy function which takes as input an underlying latent vector and input image. Our energy function is a residual network. To condition the residual network on an underlying latent vector, processing of the image at each individual residual block is modulated using a set of conditional gains and biases (FILM) inferred from the latent vector.
>
> In particular, given an intermediate feature x input into a residual network, we map an underlying latent vector to two different sets of conditional gains $g_1, g_2$ and biases $b_1, b_2$ using a single linear layer. We then modulate the outputs of the first convolutional layer in the residual block, $x = conv_1(x) * g_1 + b_1$, by regressed conditional gains and biases. We apply a non-linearity to this output x, and then further modulate the outputs of the second convolutional layer in residual $x = conv_2(x) * g_2 + b_2$ by the second set of regressed conditional gains and biases. We apply this modulation to every single residual layer in the energy function, and detail the number of individual residual layers used to train COMET in Appendix A.3.
>
> When doing unsupervised object level decomposition, we infer individual latents for each underlying object using an encoder on an input image with spatial attention. In particular, given an input image, we encode the image to a lower dimension L x L grid with C channels using three 3 residual blocks. We then initialize a LSTM network to iteratively extract features from this low dimensional grid. In particular, to infer the latents for a particular component, we execute a LSTM, and concatenate the current cell state of the LSTM with the C dimension feature vector at each grid location. We map these resultant features to an attention score per grid location utilizing a 2 layer MLP on features at each grid location.  We compute softmax weights over each grid location through attention scores, and use attention scores to aggregate the LxL grid to a single C dimensional vector. The resultant C dimensional vector is input to a LSTM, which predicts the latent for this component.

---

### Official Review · Reviewer_7WaM · 2021-07-16

**Rating:** 6
**Confidence:** 4

**Summary:**

Authors propose a model that decomposes images in terms of a set of 'components', which are energy terms whose minimisation gives rise to the observed pixels. Unlike previous works on compositional EBMs, the different components are learnt/discovered without supervision, just by learning to reconstruct a dataset of images. Results on various datasets show the method has significant potential, though there is considerable room for improvement. In particular, the method discovers concepts corresponding to different objects in multi-object scenes, and different attributes of faces.

**Limitations And Societal Impact:**

Discussion is brief but adequate.

**Main Review:**

Model & theory:

++ The model is novel and elegant, and a significant extension of earlier works on energy composition

++ The approach is clearly motivated and carefully described

++ The theoretical results are reassuring (although unsurprising)

-- It should be made clearer early on that there is no probabilistic interpretation attributed to the energy functions here (in contrast to typical EBMs)

-- The need for deciding a priori which component 'slots' will be local and which will be global is awkward, and somewhat limiting where this is unknown

-- 171: the analysis here does not seem so relevant -- surely what is important in practice is expressivity when the argmin is calculated only approximately

Experiments:

++ Decomposition & recombination results where components correspond to objects (CLEVR & Tetrominos) are qualitatively good

++ Quantitative disentanglement results are better than beta-VAE

-- Segmentation is measured using ARI, which is rather insensitive to certain failure cases. Mean segmentation covering should be given instead (as in most other works on object-centric modelling)

-- Disentanglement & recombination where components correspond to attributes (faces & KITTI) are not so good qualitatively

-- Reconstructions are not very high fidelity compared with other recent models

-- 299: "cross-dataset" seems rather a strong designation, given the similarity of the datasets in question

---
## Post-rebuttal

The authors have addressed my concerns sufficiently in their response, and also (in my opinion) those of the other reviewers. Therefore, I still favor acceptance, provided the promised evaluations etc. are added in the camera-ready.

**Time Spent Reviewing:**

2

---

> ### Author Response · Authors · 2021-08-10
> **Reviewer 7WaM Response**
>
> Thank you for your detailed comments and feedback; we appreciate the time and attention you gave to reviewing our paper. We address concerns for additional evaluation and overall clarifications of the text below.  Please let us know if you have any additional questions, we are happy to clarify or provide additional experiments.
>
> ------------------------
> **Q1) No Probabilistic Interpretation.**
> We will clarify in Section 1 and 3 that there is no probabilistic interpretation attributed to the energy functions.
>
> **Q2) Specification of Global and Local Slots.**
>  We believe that the ability to specify which slots are global and which slots are local is actually beneficial and enables us to obtain a controllable decomposition of a scene. Such an ability enables us to choose and control, given an image, exactly the number of local factors to obtain and the number of global factors, depending on our use case/application. This allows us to decompose a scene into, e.g. two global and two local factors, or through an alternative specification, into all local factors of variation, or any other type of decomposition we desire.
>
> **Q3) Utility of Theoretical Results on Energy Decomposition.**
> Our theoretical results on energy decomposition establish that given exact optimization of an energy function, there are significant computational benefits towards utilizing energy functions, as the optimization procedure removes some of the representational burden of the energy function. While in practice we utilize approximate optimization on an energy function, we believe a similar intuition holds, where the approximate optimization procedure alleviates the representational burden of the associated energy function also. We will clarify this in Section 4. Part of our performance gains may be due to increased expressivity, and we provide some light analysis of this in Remark 1 of Section 4. We will add further discussion regarding this.
>
> **Q4) Segmentation Metrics.**
> We have evaluated unsupervised object segmentation utilizing mean segmentation covering. We find the MONET obtains a mean segmentation covering of 0.701, while COMET obtains a mean segmentation covering of 0.713, with both methods performing relatively similarly.  We will add this to the results in Section 5.2.
>
> **Q5) Image Quality.**
> See Section 1.3 of the General Response.
>
> **Q6) Cross Dataset Recombination.**
> See Section 1.2 of the General Response.

---

### Official Review · Reviewer_rFPF · 2021-08-02

**Rating:** 6
**Confidence:** 3

**Summary:**

The paper proposes a novel unsupervised learning method called COMET that discovers global and local factors of variation in image datasets by representing these factors as energy functions. The experiments show that the proposed approach performs better than beta-VAE in terms of disentanglement evaluation. Qualitative examples on various datasets show that the proposed approach captures global and local object-based factors of variation and is able to generate images from various combinations of such factors well.

**Ethical Concerns:**

I do not think that there are any ethical concerns related to this work.

**Limitations And Societal Impact:**

Yes, the authors discuss their work’s limitations and societal impact.

**Main Review:**

Strengths:
- The proposed approach allows combination of factors of variation in a non-linear manner, enabling more complex relationships to be modeled.
- The authors show how the proposed approach is more expressive than the previous work that combines segmentation masks.
- Experiments on multiple datasets show the generalization capacity of the proposed approach.
- The paper is well written.


Weaknesses:
- Can the authors give examples where beta-VAE “is unable to compose multiple instances of the same component or a larger number of components than seen during training”?
- The evaluation is lacking:
    - How do the authors tag the inferred energy functions with semantic concepts? Manual inspection by the authors? In many cases, they do not seem intuitive — how did the authors determine that the first energy function is camera position? Looking at Figure 3, the third energy function seems to be focussing on blue-colored objects and the fourth energy function seems to be highlighting the complete image. Like other disentangled representation learning works such as beta-VAE, can they generate sets of examples where only this factor of variation is changing while everything else remains the same? Do they observe the camera position changing clearly? Same study for all the factors of variation that have been tagged with semantic concepts.
    - The proposed approach has only been compared against beta-VAE? What’s the reasoning behind this? Why not with MONET? Why not with other approaches mentioned in the related work as well as other methods that learn disentangled representations such as InfoGAN? Although [12] requires concept labels, it would still be informative to compare how energy functions perform in the presence vs absence of concept labels.
    - The shown beta-VAE reconstructions look odd. The reconstructions for CelebA-HQ seem too reddish. What implementation of beta-VAE is being used?
    - Factor recombination can also be done with other models such as beta-VAE, right? How do the results shown in Figures 4 and 6 look for beta-VAE?
    - Figure 5: I don’t see any change by adding the “Lighting” energy function, what should be noticed here? Can the authors show the generations when these energy functions are added in the opposite order — lighting, then floor and then objects?
    - Table 1: what is the variance across different seeds? Also, please change the answer to 3(c) in the checklist to [No] since that’s clearly not true.
    - The resolution of images is so low that it’s hard to make out the shapes in CLEVR images and understand the semantics of KITTI images. Why is that? Can the resolution be increased to mitigate these problems?
    - How does the proposed approach compare to other methods in terms of computational time/resources?
    - What is the use of cross-dataset recombinations? What might be an example where this might be needed?
- Minor:
    - Since the code is currently not provided, the answer to 3(a) in the checklist should be [No].


------------------------------------------------------------------------------------------------------------------------------------------------------------------------------------------
------------------------------------------------------------------------------------------------------------------------------------------------------------------------------------------
I thank the authors for their response to my questions/concerns and the additional qualitative and quantitative results. The response answers most of my concerns. Therefore, I still recommend acceptance for the paper. However, I still have some concerns that, when fixed, would make the paper much stronger in my opinion --

1. There is something wrong with Beta-VAE implementations used in the paper, resulting in weird qualitative results and perhaps affecting quantitative comparisons.
2. Many qualitative results are hard to understand and appreciate because of the low resolution of images.
3. The interpretability of the decomposition of natural images (Falcor3D, CelebA-HQ, KITTI) is a bit subjective.

**Time Spent Reviewing:**

4

---

> ### Author Response · Authors · 2021-08-10
> **Reviewer rFPF Response**
>
> Thank you for your detailed comments and feedback; we appreciate the time and attention you gave to reviewing our paper. We have addressed concerns about evaluation in the comments below, and will further update the checklist.  Please let us know if you have any additional questions -- we are happy to clarify or provide additional experiments.
>
> ------------------------
> **Q1) Examples of Beta-VAEs Unable to Compose Across Multiple Instances.**
> An example of the beta-VAE being unable to compose across multiple instances of the same component can be found when a scene contains multiple objects such as those considered in Section 5.2.  The Beta-VAE formulation requires that each object in the scene occupy a chunk of an underlying latent space. For two different images, the same chunk of the latent space will then describe two different objects in each scene. However, if we wish to construct a scene with all objects from both images, we must now construct a latent space where the same chunk of latent space describes both one object in one scene and a second object in a second scene. There is no way to construct such a latent chunk, since an interpolation in the latent chunk will instead construct a different object. Our approach, as shown in Figure 7, is able to compose 8 objects across two separate images together. We will add a discussion about this in Section 5.2 of the paper.
>
> **Q2) Inferred Semantic Concepts.**
>  The semantic concepts represented by each inferred component were determined by manually recombining each energy function with separate ones from other images. From visual inspection, the first energy function (corresponding to camera position) in Figure 3, can be seen by attending to the edges of the objects in the image, but we agree that the underlying visualization can be clearer. We will improve the visual in our revision.
>
> In Figure 4 of the main paper, we provide examples where a single component is recombined across two images. Directly varying a single component, similar to what is done in beta-VAE, is more difficult for COMET due to the fact that underlying components are not directly embedded in a continuous Gaussian latent space. However, on the Falcor3D dataset, we are able to visualize this variation across each single component, as the dataset contains renderings of all discrete combinations of underlying factors, enabling us to infer latents representing variation of each factor. In https://ibb.co/f8bbwhr, we illustrate on the left side, variations of a single component representing lighting intensity, lighting direction, and camera position (for camera position, we use energy functions for both camera position and floor position), when combined with fixed energy functions for the remaining components. On the right side, we show ground truth renderings. We find that our renderings closely match ground truth renderings.
>
> **Q3) Reproducibility of Seeds.**
> In our original results in Table 1, each row corresponded to a different random seed. In https://ibb.co/QngJT61, we have rerun all results in Table 1 across 3 separate seeds, and have reported the mean and standard deviation. We thank the reviewer for this suggestion, and will add this to the paper.
>
> **Q4) Baseline Comparisons.**
>  We choose to compare to Beta-VAE for disentanglement evaluation, as the method was proposed for disentanglement, with many subsequent works building upon it [5, 8, 24, 36], and has been shown to perform comparably to recent techniques (see https://arxiv.org/abs/1811.12359). We will clarify this choice in the revision.
>
> Still, we do see the value in further comparisons, and evaluate InfoGAN and MONET in https://ibb.co/QngJT61. We will update the paper with these results. Visual quality of supervised EBMs can be found in [12] -- our approach understandably generates blurrier images than a supervised EBM.
>
> **Q5) CLEVR Visualization.**
> To more clearly illustrate the effect of adding lighting to CLEVR generation, we consider adding factors in the order of floor, lighting and object components and visualize the result in https://ibb.co/NWQj3qd. In our original visualization, the additional lighting component at the end of Figure 5 leads to the addition of lighting effects surrounding the cubes (can be seen clearly by magnifying the figure).
>
> **Q6) Beta-VAE Implementation and Evaluation.**
> We utilize a standard implementation of the Beta-VAE found at https://github.com/1Konny/Beta-VAE (which has been used in refereed work, see [34]; https://github.com/bethgelab/slow_disentanglement). We are unsure why the images have reddish hinge.To further validate this, we run an additional experiment using https://github.com/matthew-liu/beta-vae and find similar results of images having a reddish hinge, albeit also being blurry. We attach images of different reconstructions at https://ibb.co/FXttM3c. We further quantitatively evaluate the disentanglement results of this second implementation of Beta-VAE, and find that it obtains similar performance (see https://ibb.co/xJTH8Ts ). Note the high variance in the results of Beta-VAE, which we do not observe with COMET. We will add a discussion of this in Section 5.1 of the paper.
>
> In terms of utilizing the Beta-VAE for Figure 4 and 6, we found it difficult to determine the underlying latent dimensions representing corresponding semantic factors in beta-VAE.  We note that intermediate interpolations also exhibit similarly poor image quality.
>
> **Q7) Image Quality.**
> See Section 1.3 of General Response.
>
> **Q8) Computational Resources.**
> Our approach, since it requires backpropagation through the gradients of the neural network, is slower to train than baseline models. While baseline methods are run over the course of approximately 6 hours, COMET models are trained over a period of approximately 1 day. However, we believe such increases in the computational power needed for training is justified by the increase in performance of our approach over baselines in terms of disentanglement. Furthermore, our approach provides fundamentally new capabilities in cross dataset recombination. We will add said details to the appendix of the paper.
>
> **Q9) Importance of Cross Dataset Recombination.**
>  Typically, in deep learning, we assume access to a large pre-gathered dataset on which models are trained. In practice, in the real world, we experience a continuous stream of new data. To effectively learn in the real world, we need to construct deep learning models which can combine knowledge learned from one experience with that learned from another experience. We see our model’s ability in cross dataset recombination as an initial step towards this broader goal of composing visual concepts across different experiences. We will add more motivation for this in the introduction.

---

### Author Response · Authors · 2021-08-10
**General Response**

We thank reviewers for their detailed comments and feedback. Reviewers agreed that the paper was novel and interesting (7WaM,  2JFE) and well written (rFPF, 7WaM, 2JFE) and were unanimously positive about the paper. Reviewers had some shared concerns which we detail below and further address in the individual author reply. Please let us know if there are any additional results or textual clarifications that we can provide.

--------------------------
### 1.1 Evaluation

Reviewers had some shared concerns about the underlying evaluation of the method. We list the additional experiments run below -- please see individual responses for detailed analysis of these results. Both reviewers rFPF and 2JFE had concerns about the disentanglement of individual components inferred by COMET. To better showcase the individual factors captured by global factor decomposition, we present results where we only vary a single energy function for a factor of variation that is either camera position, lighting direction, or lighting intensity in https://ibb.co/f8bbwhr, where the right side shows ground truth generations with regards to the same factor. To further understand the underlying disentanglement of COMET,  we further run all disentanglement results across 3 separate seeds, with the additional baselines of InfoGAN and MONET in https://ibb.co/QngJT61. We also provide a explanation of the underlying reason for our disentanglement results in reviewer 2JFE's personal response, and show that a t-SNE plot of embeddings of separate components are separated https://ibb.co/mc6ZvZx.

Reviewer 7WaM had concerns about the evaluation of unsupervised object discovery,  so we add additional comparisons of methods using mean segmentation coverage metrics. Reviewer rFPF had further concerns about our underlying Beta-VAE implementation so we compare disentanglement metrics with a second implementation of Beta-VAE in https://ibb.co/xJTH8Ts.

--------------------------
### 1.2 Cross-Dataset Recombination
Reviewers rFPF and 7WaM also had individual questions about the cross-dataset recombination setting we consider.  Reviewer rFPF asked about the underlying usefulness of cross-dataset recombination, while Reviewer 7WaM questioned the simple nature in which cross-dataset recombination is done. We believe that for deep learning to be deployed in the real world, it is crucial that models be able to combine in a cross-dataset manner, as when separate models are trained on separate experiences, the question on how to integrate models learned from distinct data sources needs to be addressed. Our method provides a possible approach to tackle this problem.

In terms of dataset distinctiveness in our cross-dataset setting, while we agree that the underlying datasets through which we recombine COMET models are similar, we stress that our results show a fundamentally new capability which cannot be done by prior methods. In particular, there is no clear manner in which we may combine the two separate latent spaces obtained from beta-VAE on two different datasets. Similarly, it is hard to compose different segmentation masks inferred by MONET on two different datasets, especially if they are overlapping with each other.

--------------------------
### 1.3 Image Resolution
All reviewers has comments about the overall generation quality of our method. While it's desirable to have good generation quality, we believe our existing results validate that our approach can capture global/local factors of variation, and can combine across datasets. Furthermore, we believe that our approach outperforms existing work in disentanglement and unsupervised object decomposition on the datasets we considered, as illustrated by the reconstructions at https://ibb.co/FXttM3c. Notably, recent such methods often evaluate at 64x64 resolution, and on simpler datasets than those considered in our paper, i.e. dSprites [23].  Nevertheless, we achieve higher generation quality in COMET by training a model at 128x128 resolution on CelebA-HQ images at https://ibb.co/nMxRZx6 . While we agree that some recent works have shown higher resolution results, we note that many such works rely on large pretrained models, such as StyleGANv2, which are trained at a larger computational scale than we unfortunately have access to. We will discuss this in the related work as well as in the limitations section of the paper.

--------------------------
### Conclusion
Each individual reviewer further has their own personalized questions about the underlying text and statements of the paper. We address each of these concerns in the individual author comments, and also provide detailed discussion of evaluation results in the individual reviewer comments. Please let us know if there are any additional questions or requests for additional experiments.

---

### Author Response · Authors · 2021-08-24
**Rebuttal Response**

Dear Reviewers,

Thank you for all the time you spent reviewing the paper and the thoughtful comments given. We were wondering if you wouldn't mind taking a look at our updated results. We have added additional quantitative and qualitative evaluation as requested by each individual reviewer.

Thanks,
Paper Authors

---

### Decision · Program_Chairs · 2021-09-27

**Decision:**

Accept (Poster)

**Comment:**

There is a reviewer consensus that the paper is just above the bar. I concur with that.